# Rural Development from a Gender Perspective: The Case of Women Farmers in Southern Spain

Jaime De Pablo Valenciano [ID], Juan Milán-García *, Juan Uribe-Toril [ID] and María Angustias Guerrero-Villalba

Area of Applied Economics, Business and Economics Department, University of Almeria, 04120 Almería, Spain; jdepablo@ual.es (J.D.P.V.); juribe@ual.es (J.U.-T.); maguerre@ual.es (M.A.G.-V.)
* Correspondence: jmg483@ual.es

**Abstract:** This article analyses the contribution to local development by women workers in the fruit- and vegetable-handling sector in Almería (Spain) over the last five years (2015–2019). It is a continuation of research carried out during the period 2000–2014. Using data collected through surveys and focus groups, the aim is to ascertain if the results obtained in this analysis meet the condition of sustainability, i.e., whether the improvement in working women's quality of life has been maintained over time, and whether these beneficial effects have multiplied. The results show that women workers in the fruit- and vegetable-handling sector are satisfied with their jobs and with the company they are working for. The existence of fixed-discontinuous employment contracts facilitates greater flexibility for women in terms of balancing work and family life. This main contribution of this study lies in extrapolating the sustainability of a local development model in regard to other initiatives that aim to increase women's empowerment in the labour market.

**Keywords:** women empowerment; sustainable local development; food-handling sector; horticultural sector



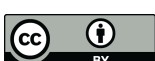

## 1. Introduction

The role of women in the agricultural sector has been on the rise in recent years. In some countries, such as Nigeria and Pakistan, women are involved in subsistence agricultural production. In addition to subsistence farming, they may also receive income from the formal and informal sectors, depending on their means of living and poverty level. In Africa, some women are willing to engage in collecting forest resources to support their families. However, inequality between men and women remains evident in some parts of Africa. This inequality reduces women's opportunities to improve their family's situation and their household economy. Additionally, women play a crucial role in food security [1].

In some regions, such as Western Europe [2], which also have a high concentration of migrant workers, there is a need for women's participation in the agricultural and industrial sectors. In other regions, such as the African continent, social pressure from ethnic groups makes it difficult for women to engage in economic activities [3].

The analysis presented in this study is based on the province of Almería, located in the south of Spain (Figure 1). This territory was chosen because Almería is one of the leading centres of fruit and vegetable production in Spain, and in Europe in general. In fact, it is known as the orchard of Europe [4]. In this region, agriculture is concentrated in three main areas: La Cañada, Campo de Níjar, and Campo de Dalías.

In terms of freshness, fruits and vegetables in Almería are considered a touchstone in Spain. The area is also considered a global reference in this sector [4,5].

The commercial system in Almería is structured at a business level as follows:

- Establishments that sell at origin (*alhóndigas*). These take the legal form of public limited companies.
- Establishments that sell in destination markets (usually cooperatives or agricultural processing companies (APCs)).

- The San Isidro Agricultural Cooperative (SAIC), a cooperative that sells at origin

For the companies that sell at origin (*alhóndigas* and SAIC), the supply from the agricultural producers and the demand from the commission agents compete with sales being made through reverse (or Dutch) auctions. In turn, these brokers work for third parties, including the cooperatives themselves, APC in Almería and other provinces, and the consignees of central markets and purchasing centres of large supermarkets. While the reverse auction system originated in the Netherlands, it is no longer used in that country. However, it is still used in Belgium (known as *veiling*) and France (known as *cadran*), particularly in the Brittany region.

Companies selling at destination (cooperatives and APC) operate directly with the central purchasing agencies of supermarkets.

Figure 2 presents an overview of the main aspects of the value chain that characterise fruit- and vegetable-handling activities. As can be seen, the activities begin with the entry of products into the warehouse followed by transportation and preparation for wholesale or retail sale.

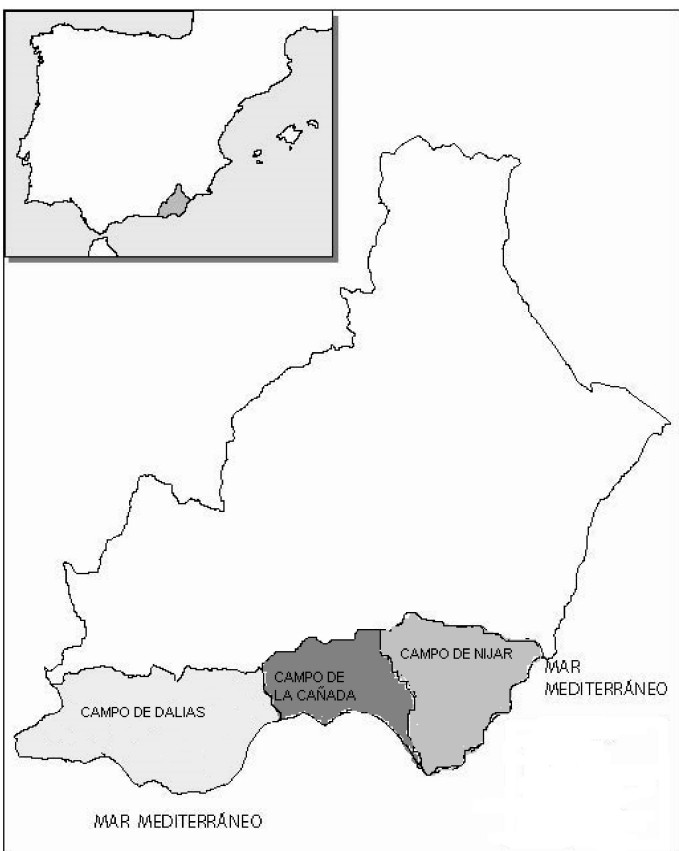

**Figure 1.** Almería, polled areas. Compiled by the authors.

This paper analyses the contribution to local development by women workers in the fruit- and vegetable-handling sector in the province of Almería (Spain) over the last five years (2015–2019). It is the continuation of research carried out during the period 2000–2014, which showed that the presence of women in this sector increased their empowerment [6]. Therefore, this paper aims to evaluate whether this process of empowerment has been consolidated in recent years, guaranteeing a sustainable process of local development that permeates all sectors of the population. In addition, it evaluates the safety of women participating in the handling of fruit and vegetables, a fundamental aspect in guaranteeing this sustainability.

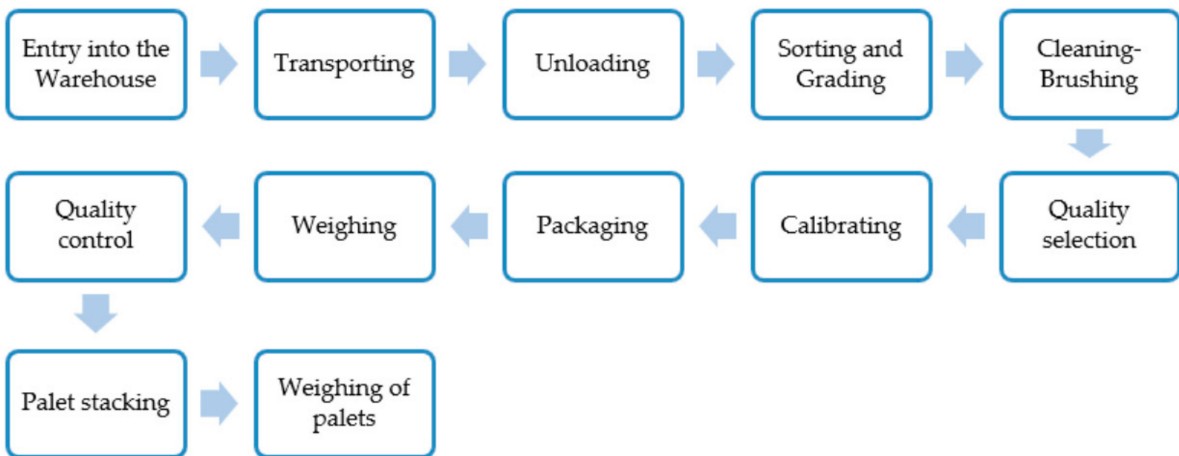

**Figure 2.** Handling process.

## 2. Understanding Rural Development

Throughout history, women's employment has been characterised by a low presence in the production chain and restrictive working conditions that have hindered their ability to balance work and family life [7]. This situation is further complicated if women's backgrounds are taken into account, as prejudices further raise barriers to women's access to the labour market [8] and the sector in which they work [9].

In recent years, general awareness of women's participation in the labour market has grown [10], highlighting the need to integrate a gender perspective into any business, as well as into social and cultural projects [11]. However, the data show that women still occupy a secondary position in the work environment, suffering from phenomena such as the well-known glass ceiling, which refers to the difficulty women have accessing high-level positions within companies [12].

At the international level, differences between countries regarding women's participation in the labour market are significant in both quantity and quality [13]. There are also differences between urban and rural environments, with the latter being the most difficult for women [14]. However, some research papers state that when men move to urban areas, it leads to a gradual feminisation of agriculture [15]. In an African context, the difference between women in rural and urban areas is significant [16]. Resources for subsistence farming in rural areas are both insufficient and lack quality. According to [17], life is neither satisfying nor decent for millions of women in rural Nigeria.

In comparison, women's participation in the fruit- and vegetable-handling sector is higher than in other primary-sector activities (fishing, livestock) or other industries (construction, technology). This is due to the characteristics of the food-handling activities, which require greater flexibility and agility [18].

The empowerment of women in the workplace is manifested in their satisfaction with the tasks they perform, with the environment they experience during the workday, as well as with the company that hires them [19]. Scientific literature has addressed the satisfaction of women workers in the fruit- and vegetable-handling sector. Researchers highlight aspects that affect satisfaction, such as the devaluation of women's work [20] or the need to improve working conditions in order to increase the satisfaction of working women, both in terms of physical and mental health [21–23].

The concept of rural development refers to the improvement of the quality of life of rural citizens through initiatives based on endogenous resources that are economically and environmentally sustainable in the long term [24]. The fruit and vegetable sector has been key to the rural development of the Almería area [4], as well as other areas of Spain, such as Huelva [25].

In rural areas, the role of women is key to developing innovative rural entrepreneurship initiatives [26]. Therefore, in Spain, a series of strategies have been articulated to

increase the empowerment of women in rural environments, such as the Strategy for Rural Modernisation and Diversification or the Strategic Plan for Equal Opportunities [27].

## 3. Materials and Methods

Through a systematisation of experiences, the papers referred to in the previous section were analysed. This methodology was chosen because it is considered an innovative and creative approach to learning from a multitude of different experiences [28]. Furthermore, a survey was developed based on expert interviews.

### 3.1. Phases of the Study

- First phase: Based on the following goals, we tested our initial hypothesis:
  - Detect whether certain situations of economic growth and business dynamics (organisational and technological changes) influence the greater participation of women in the workplace.
  - Understand how the processes of economic growth and business dynamics in the sector have affected working women, particularly women in positions of responsibility.

To carry out this research, we focussed on data collection and interpretation of information involving trade unions and workers in a number of companies.

- Second phase: Based on the information collected, a questionnaire was developed for the companies, combining personal questions and specific details related to the sector. The research was completed following the subsequent steps:
  - A series of surveys and interviews were used to cross-reference the information from the handling companies.
  - In-depth interviews with men and women working in the horticultural sector were conducted, including interviews with social agents and institutions.
  - Technological changes in the local production system were analysed.
  - The effects of innovation on working women in the horticultural sector were evaluated.
- Third phase: Diagnosing study.

The qualitative part of the study was based on in-depth interviews carried out within the sector and was accompanied by hypothesis testing and analyses of the data obtained from the survey conducted in the second phase of the study.

### 3.2. Companies Studied

Most companies in the sector were established before 1985, a year which is considered the starting point of the development of horticultural companies in Almería. The early 1980s saw the establishment of cooperatives and APCs. In recent years, there has been an increase in the number of *alhóndigas* (places used as both warehouses and marketplaces), while the same has not happened in social economy enterprises. It is important to consider the age of the companies to understand the evolution of working conditions and to verify if feminisation has been observed in certain positions. The 25 companies in the fruit-and-vegetable handling sector in Almería with the highest turnover during the period 2014–2019 were selected since they employ a significant part of the female workforce. All companies had a turnover of more than 75 million euros per year, with two of them exceeding 230 million euros per year.

## 4. Results

### 4.1. Typology of Workers and Shareholders

The presence of women in positions of responsibility in fruit and vegetable companies has been debated over the past few years. Compared to 2014, there are now significantly more social economy companies and *alhóndigas* (Table 1), particularly in sectors such as field technicians, marketing, occupational-risk prevention, and quality-control departments. However, in high-responsibility positions, such as management, the number is insignificant.

**Table 1.** Shareholders.

| | Social Economy Enterprises | | Alhóndigas (Auctions) | |
|---|---|---|---|---|
| | **2014** | **2019** | **2014** | **2019** |
| Total number of shareholders | 17.051 | 18.067 | 425 | 502 |
| % Women | 6.82% | 9.65% | 8.2% | 10.02% |

Source: Compiled by the authors.

*4.2. Age and Nationality of Women in the Food-Handling Sector*

The data analyses reveal that, in the food-handling sector, over 90% of workers are women. Most of them are Spanish and married (Table 2). The results obtained from the partial surveys do not evince significant differences among the data by area.

**Table 2.** Horticultural areas studied.

| Questions | Campo Dalias | La Cañada | Campo de Níjar |
|---|---|---|---|
| Age (%) | | | |
| <20 | 4 | 5 | 0 |
| 20–30 | 27 | 32 | 39 |
| 30–40 | 28 | 37 | 27 |
| >40 | 41 | 26 | 34 |
| Nationality (%) | | | |
| Spanish | 76 | 83 | 78 |
| Moroccan | 7 | | |
| Romanian | 4 | | |
| Bulgarian | | 9 | |
| Polish | | 3 | |
| Lithuanian | | 3 | |
| Ecuadorian | 3 | | 5.5 |
| Peruvian | 3 | | 5.5 |
| Colombian | | | 5.5 |
| Chilean | | | 5.5 |
| Others | 7 | 2 | |
| Marital Status (%Married) | 55 | 52 | 39 |

Source: Compiled by the authors.

The distribution is even across age groups, with the exception of women under 20. This is evidence that most women have been working in the sector for years.

*4.3. Education Level*

Concerning the level of education of women workers in the fruit- and vegetable-handling sector, there has been an increase in the percentage of women with secondary education in recent years, while the percentage of women with no education has decreased (Figure 3).

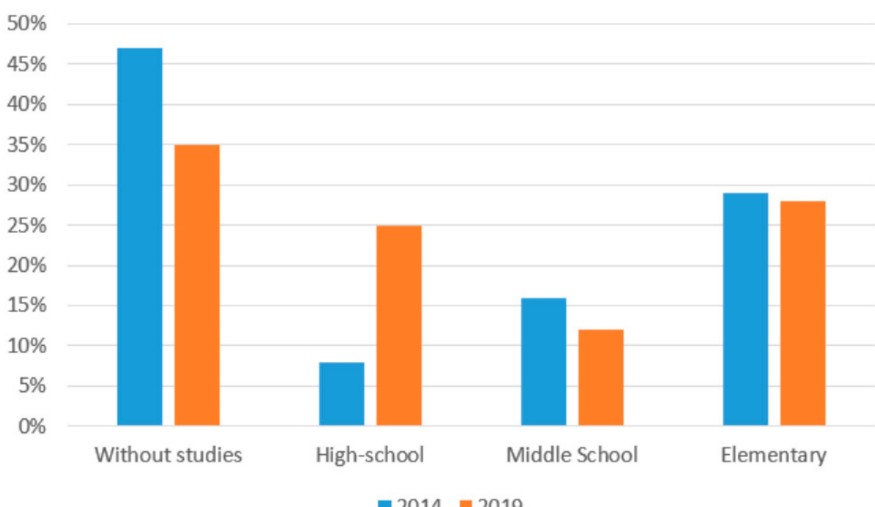

**Figure 3.** Education level of workers in fruit- and vegetable-processing companies. Source: Compiled by the authors.

*4.4. Types of Contract*

The taxonomy of professionals can be found in the collective employment agreement (ECA). There are different professional subcategories charged with handling and packaging in the fruit, vegetable, and flower sectors. The taxonomy has been designed according to legally-established criteria and based on contractual relation, i.e., according to the type of working contract chosen and formalised by the parties.

- Fixed/Permanent: Those that work for the business on a regular and continual basis.
- Fixed-discontinuous: Those that carry out regular, discontinuous work, which is seasonal or intermittent, for horticultural campaigns.
- Casual/Temporary workers: Those with an indefinite relationship with the business, whose employment is conditioned by market circumstances or workload, including during regular campaigns. The maximum length of these contracts is about nine months over a twelve-month period, calculated from the moment the relationship is initiated.
- Interims: Those that are contracted as substitutions of fixed-term or fixed-discontinuous staff.
- Single job or service: These positions are aimed at not overfilling the staff with fixed-discontinuous workers. Workers are not guaranteed employment during the whole season, so the parties agree that contracts for a single job or service can be completed for typical campaign tasks. However, this kind of contract should not constitute more than 50 percent of the staff.

The survey shows that in the last five years, the number of fixed-term and fixed-discontinuous contracts has vastly increased, giving women the security that comes with stable income streams (Figure 4).

As discussed in the previous study, the number of working hours per day ranges from six to ten, depending on the seasonal activity (Table 3).

**Table 3.** Hours of work per day.

| Campaign/Hours | <5 | 6 to 9 | 10 to 15 | <15 |
|---|---|---|---|---|
| High Activity | 4% | 42% | 52% | 2% |
| Low Activity | 71% | 29% | 0% | 0% |

Source: [4].

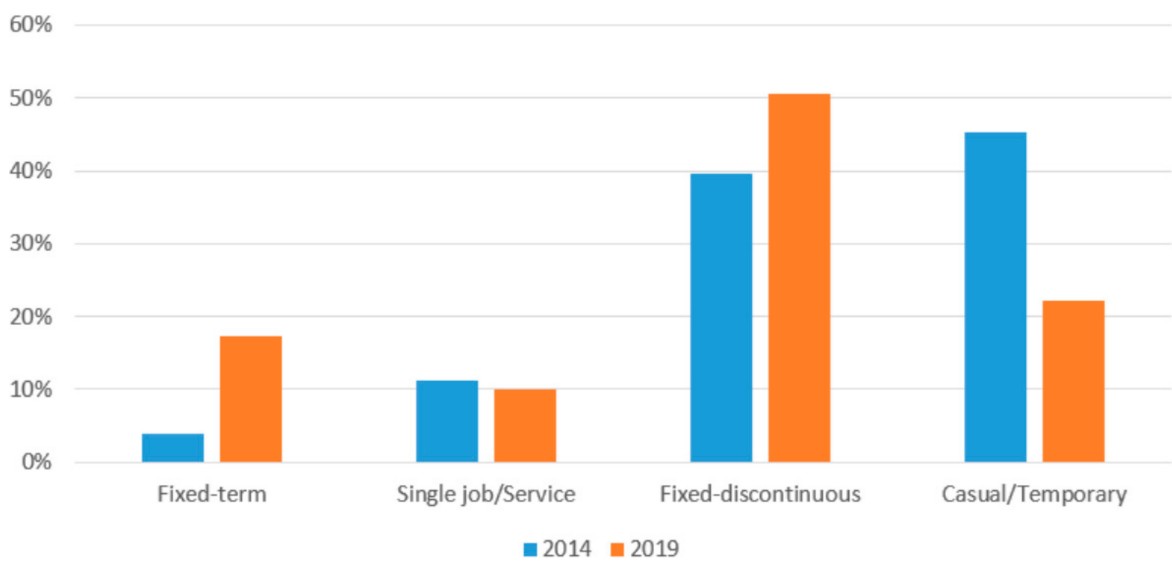

**Figure 4.** Types of contracts. Source: Compiled by the authors.

### 4.5. Impact of Workplace Accidents and Occupational Hazards in the Food-Handling Industry

At this juncture, it is essential to understand how company size can influence the number of accidents. Thus, information was collected from a sample set of 25 companies in the sector (calculating the average over the years 2019 and 2018), where NA = number of accidents, TC = tons sold, and NW = number of workers. First, asimple correlation analysis of the variables (first panel of Table 4) was conducted. As shown, all variables are closely related. It is evident that the number of accidents is correlated with the trade total (r = 0.78) and the size of the business. This is to be expected, although it also indicates that a bigger company demands a higher level of organisation and greater control in order to reduce the NA. However, this has not been verified, as can be confirmed by looking at the correlation coefficient between NW and NA (0.94). Using ratios helps deepen the analysis. The *productivity index* (PI = TC/NW) and *incidence rate* (IR = NA/NW) are two variables used in the analysis. One might think that the company with the highest productivity (PI) would use the resources more intensively, thus affecting a larger NA, a hypothesis that is not supported by the data.

**Table 4.** Correlation between variables.

|      | **NW**  | **NA**  | **TC**  | **IR**  | **PI**   |
|------|---------|---------|---------|---------|----------|
| NW   |         | 0.94    | 0.63    | 0.14    | −0.25    |
| NA   | 0.94    |         | 0.78    | 0.36    | −0.22    |
| TC   | 0.63    | 0.78    |         | 0.41    | 0.06     |
| II   | 0.14    | 0.36    | 0.41    |         | −0.02    |
| IP   | −0.25   | −0.22   | 0.06    | −0.02   |          |

Sample of 25 companies. Source: Compiled by the authors.

The companies studied employ approximately 40% of the total number of women who work in the food-handling sector of the three main areas: La Cañada, Campo de Níjar, and Campo de Dalías.

According to data from the Andalusian Regional Government's occupational-risk prevention service, there has been a considerable decrease in the incidence rate over the last five years (Table 5).

**Table 5.** Evolution of the magnitudes of incidence rates.

| Year | Number of Firms [1] | Average Number of Employees per Firm (1) | Average Number of Accidents per Firm (2) | Incidence Index [(2)/(1)] $\times$ 100 | Lost Days per Company |
|---|---|---|---|---|---|
| 2014 | 56 | 128 | 23 | 17.9 | 364 |
| 2015 | 60 | 141 | 25 | 17.7 | 353 |
| 2016 | 59 | 138 | 24 | 17.4 | 335 |
| 2017 | 58 | 135 | 22 | 16.3 | 330 |
| 2018 | 57 | 130 | 21 | 16.2 | 325 |
| 2019 | 60 | 142 | 23 | 16.2 | 325 |
| Average (2015–2019) | 58.8 | 137.2 | 23 | 16.76 | 333.6 |

[1] The firms represented are all social enterprises and fall into two categories: cooperatives and agricultural transformation societies. Accidents without sick leave are included. Source: [29].

The average accident rate for the period 2000–2014 was 26 accidents per company, and the incidence rate was almost 20%. In contrast, the period 2015–2019 was characterised by a decrease in both rates, with the average number of accidents per company falling to 23 and the average rate of incidents decreasing to 16.76%. Therefore, the period 2015–2019 evinced the efforts made to increase the occupational safety of workers.

An analysis of women's empowerment and job satisfaction should include an understanding of the degree of risk involved in their work. The workplace incidents with the highest index of danger are overexertion and falling objects. (Table 6).

**Table 6.** Main position and risk analysis.

| Position | Main Risks |
|---|---|
| Junior Warehouse Worker | Using work equipment:<br>- Falling boxes<br>- Removal of boxes<br>- Manual pallet removal. |
| Forklift Operator | High workload:<br>- Forklift<br>- Truck (falls, overexertion, and crashes). |
| Handling and Packaging Workers | Quantity and monotony<br>- Overexertion<br>- Falls<br>- Bumping into mobile objects<br>- Machine blockages. |

Most of the injuries suffered by women are specific to the lower and upper limbs (Figure 5).

To understand the recent improvements in the workplace, it is important to consider that the fruit- and vegetable-handling companies are applying the Hazard Analysis and Critical Control Points (HACCP), as well as occupational-risk prevention. This widespread use is due, among other reasons, to the market (customers), which demands their application. It is an excellent example of private initiatives demanding stricter limits that have surpassed public regulatory capacities.

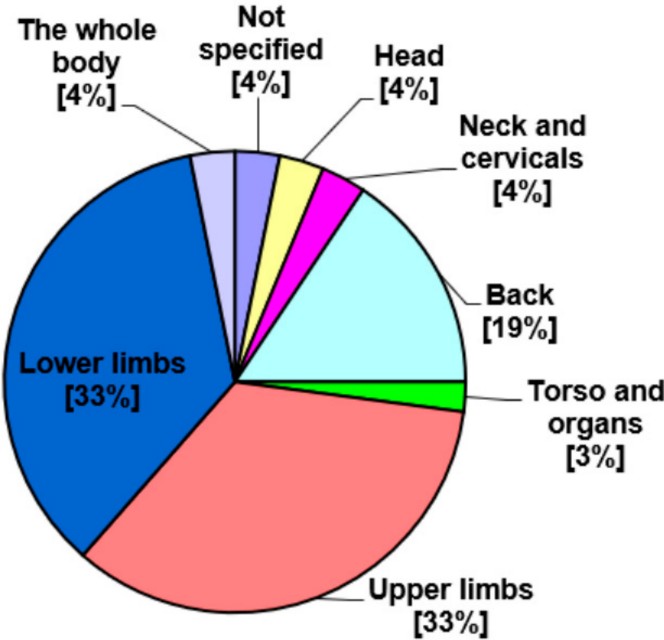

**Figure 5.** Accidents according to the anatomical region of the body affected, 2019. Source: Compiled by the authors.

### 4.6. Level of Job Satisfaction

A total of 86% of respondents replied that they did not suffer any discrimination in their working life, compared to 14% who replied that they experienced discrimination during the period.

According to the respondents, the leading causes of discrimination were the following:

- machismo (women consider that men work less but earn more than them)
- pregnancy
- between colleagues (temporary vs. indefinite term workers and between workers in different family situations).

In the final year studied, most women rated their satisfaction level with their supervisors highly (Figures 6 and 7).

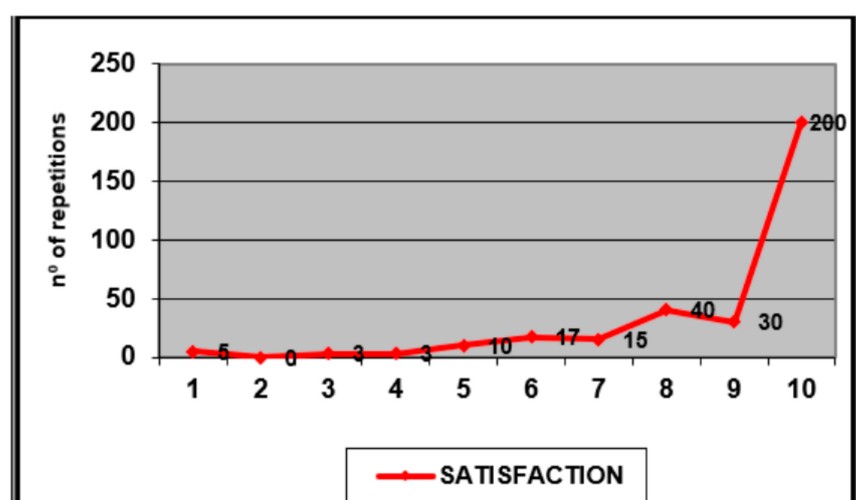

**Figure 6.** Satisfaction with women supervisors, 2019. Source: Compiled by the authors.

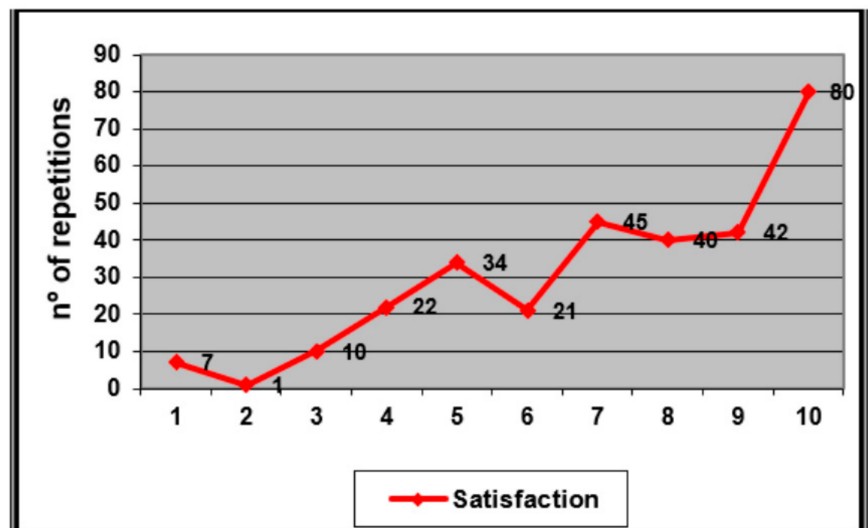

**Figure 7.** Satisfaction with men supervisors, 2019. Source: Compiled by the authors.

The women were asked about their satisfaction in the workplace using a rating scale of one to ten, where one was not at all satisfied and ten was completely satisfied. Less than 10% of women had a satisfaction level of less than five points, while more than 30% rated both male and female supervisors at ten points. However, the overall satisfaction level was higher when the supervisor was a woman. This dramatically contradicts the results obtained in the period 2000–2014, where participants reported higher levels of satisfaction with male supervisors.

## 5. Implications of Results

The results show progress in women's empowerment in the food-handling sector in the province of Almería. This has resulted in an increasing number of women in positions of responsibility within fruit and vegetable companies in Almería. However, the level of empowerment is lower when involving the women who occupy positions of responsibility [30]. Historically, the fruit and vegetable sector of Almería has been governed and managed by men, in the greenhouse-farming subsector as well as in other types of business. For example, although women work within greenhouses, the majority of these businesses are owned by men. A similar situation exists within marketing companies, where, traditionally, men have held the majority of technical, executive, and management positions, as well as serving on the board of directors. Despite this, compared to the previous period studied, this subsector can be considered to have initiated a process of feminisation [6].

Within this feminisation process, the improvement in the level of satisfaction experienced by women is relevant. It is linked to the decrease in the number of workplace accidents due to the strict application of the law regarding the prevention of occupational hazards, which has resulted in an improvement in the conditions of the workplace, the use of machinery and vehicles, and the physical and psychological characteristics of the workers (ergonomics) [31–33].

These results are in line with the situation in other countries within the European Union, which brings to light the feminine supremacy in the processing sector. Back in 2013, according to research by the European Parliament [34], in countries such as Bulgaria, France, Poland, and Lithuania, there was a greater percentage of women than men in the processing sector.

The fruit and vegetable sector has not been affected by the economic-financial crisis of 2007 [35,36] nor by the Covid-19 pandemic [37,38]. Fruit and vegetables are considered necessities and are in demand despite adversities. According to the OECD and FAO, "because food is a basic necessity, the agriculture sector is showing more resilience to

the global economic crisis than other industries." Moreover, this has meant that women working within the sector have not lost their jobs, unlike other feminised sectors such as the hotel-chambermaid sector [39,40]. Furthermore, the role of trade unions as representatives advocating for working women is fundamental [41–43].

Despite this, there is still much progress to be made to consolidate women's empowerment in all sectors and socio-economic areas [44]. First, this study has considered the role of women in the workplace from an employee perspective. However, the degree of empowerment is lower when women's entrepreneurial activities are taken into account [45]. Furthermore, as mentioned in the introduction, this low level of empowerment is accentuated in the rural environment.

The situation for women in rural areas is characterised by aspects such as the feminisation of the rural exodus, social pressure, and the sexual division of labour, which speaks of a double-discrimination phenomenon: discrimination for being a woman and for living in a rural environment. Issues related to women's empowerment, gender, family, and work-life balance are incorporated into public policies [46]. In the case of Spain, action plans such as the Rural Modernisation and Diversification Strategy or the Strategic Plan for Equal Opportunities have been implemented [27]. All of this is in line with the European Union's guidelines in relation to women in the rural world [47].

## 6. Conclusions

This study has analysed the evolution of the role of women workers in the fruit- and vegetable-handling sector to ascertain whether the progress in women's empowerment has been consolidated in recent years.

The vegetable sector in Almería is critical to both Spain and the rest of Europe and is fundamentally based on two types of companies, *alhóndigas* and social enterprises. Furthermore, most of the workers in the food-handling departments are women, which means it can be considered a feminised sector. Most of them are Spanish nationals, and there are many fixed-discontinuous employment contracts. During the agricultural season, they work, while, in the summer months, they are unemployed and receive unemployment benefits, which helps them balance their work and family life during the summer months when their children are on vacation. Thus, they receive an income every month of the year.

Concerning women workers' satisfaction in the workplace, differences can be seen before and after 2014. While in the period 2000–2014 women expressed greater satisfaction with male supervisors, there is a change of opinion in the period 2015–2019. This is evidence of greater empowerment of women workers who are no longer afraid to express more controversial opinions as they have job security in the form of a greater number of fixed and fixed-discontinuous employment contracts. In their own words: "It is not the best job I have ever had, but it's my job." The main recommendation for maintaining this positive effect on women's satisfaction and empowerment is to increase the number of women in managerial and technical positions in the handling sector.

To improve the work-life situation of these women, the most powerful tool in the medium and long term is the education of younger generations. Teaching and learning processes must include the gender perspective at all stages of education and in all areas of knowledge, from nursery school to university, and in subjects like mathematics, natural sciences, history, and foreign languages. There is ample proof that gender equality promotes sustainable local development for all sectors of society. Likewise, public policies should encourage creating mechanisms that improve the degree of family and work-life balance for women, such as the establishment of child-care centres within companies of a significant size.

Regarding future lines of research, extrapolations of this analysis to other sectors such as, for example, at the national level, the strawberry and red-fruit sector of Huelva, Spain or, at the international level, the vegetable sector in Morocco are possible. However, this sector's main limitation is the pending negotiations for the collective agreement for the

handling sector in the province of Almería, a negotiation process that has been stalled since 2018.

**Author Contributions:** Conceptualization, J.D.P.V., J.M.-G., J.U.-T. and M.A.G.-V.; methodology, J.D.P.V., J.M.-G., J.U.-T. and M.A.G.-V.; writing—original draft preparation, J.D.P.V., J.M.-G., J.U.-T. and M.A.G.-V.; writing—review and editing, J.D.P.V., J.M.-G., J.U.-T. and M.A.G.-V. All authors have read and agreed to the published version of the manuscript.

**Funding:** This research received no external funding.

**Data Availability Statement:** The data presented in this study are available on request from the corresponding author.

**Conflicts of Interest:** The authors declare no conflict of interest.

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
