# Peer review of "Rural Development from a Gender Perspective: The Case of Women Farmers in Southern Spain"

_land, doi:10.3390/land10010075_

Round 1

Reviewer 1 Report

The study aims to analyze women workers' role in the fruit and vegetable sector in Spain. In my opinion, the topic is important; however, I do not see direct relations with the journal's main scope of issues.

General comments:

Abstract:

At the end of the abstract, there is a need to add at least two sentences about the research result.

Introduction

The justification for the study is sufficient.

Literature review

There is a gap in the theory that the study contributes.

Materials and Methods

In this part, The authors did not define directly what kind of method was used.

Companies studied – page 4, line 141

Please add more information about a number of companies and the method of their selection for the study.

Results

On page 7 occurs the Authors were investigated 25 companies.  It should be presented in the materials and methods part.

Page 7, Line 241

The studied companies employ approximately 40% of the total of women who work 241 in the handling sector. – 40% of the total sample? Of the number of employed women in investigates three regions? Please be more precise.

Please add information on how table 6 and figure 5 is related to women's perspective of the study.

Page 10, Figure 6 and  7 -  what satisfaction on the level of 5 mean? what rating scale was adopted in the interview questionnaires

Discussion

This part is loosely connected with the presented results.

Conclusion

The article lacks detailed methodological information regarding the research, which does not fully understand the presented study. This part does not have information about the limitation of the study and further research ideas.

Edit comments

There is both types of citation?  Page 3, line 112. Is it necessary?

Author Response

Reviewer 1

The study aims to analyze women workers' role in the fruit and vegetable sector in Spain. In my opinion, the topic is important; however, I do not see direct relations with the journal's main scope of issues.

Thank you for the effort you have made into the review of this paper.

General comments:

Abstract:

At the end of the abstract, there is a need to add at least two sentences about the research result.

Thank you very much for your suggestion, new lines have been included in the abstract.

Introduction

The justification for the study is sufficient.

Ok.

Literature review

There is a gap in the theory that the study contributes.

Ok.

Materials and Methods

In this part, The authors did not define directly what kind of method was used.

Companies studied – page 4, line 141

Please add more information about a number of companies and the method of their selection for the study.

Thank you for your comments. We have added information about the method and the process of selection of companies (Lines 154 to 156).

Results

On page 7 occurs the Authors were investigated 25 companies.  It should be presented in the materials and methods part.

You are right, we have included it in that section (Line 154)

Page 7, Line 241

The studied companies employ approximately 40% of the total of women who work 241 in the handling sector. – 40% of the total sample? Of the number of employed women in investigates three regions? Please be more precise.

We have clarified this issue (Lines 247 to 249)

Please add information on how table 6 and figure 5 is related to women's perspective of the study.

We have added information regarding the relevance of table 6 and figure 5 in the empowerment of women (Lines 263 and 264).

Page 10, Figure 6 and 7 -  what satisfaction on the level of 5 mean? what rating scale was adopted in the interview questionnaires

We have clarified the meaning of this (Lines 301 and 302).

Discussion

This part is loosely connected with the presented results.

We have expanded the discussion part (Lines 314 to 319 and lines 322 to 327)

Conclusion

The article lacks detailed methodological information regarding the research, which does not fully understand the presented study. This part does not have information about the limitation of the study and further research ideas.

Thank you very much for your suggestion. We have included limitations and future research ideas (Lines 380 to 385).

Edit comments

There is both types of citation?  Page 3, line 112. Is it necessary?

We are sorry for the inconvenience. We have corrected it.

Reviewer 2 Report

1. Language improvement

2. Improve your conclusion point specific impact of public policy strategies in the empowerment of women 

3. Suggestions of other sector for which similar studies could be applied in the region

Author Response

Reviewer 2

  1. Language improvement

A native speaker has reviewed and translated the whole text to improve the quality of the language.

  1. Improve your conclusion point specific impact of public policy strategies in the empowerment of women 

Thank you very much for your suggestion. The conclusion section has been expanded with public policy strategies and recommendations (Lines 367 to 369 and lines 376 to 378).

  1. Suggestions of other sector for which similar studies could be applied in the región

Thank you for this comment. We have included it as future research idea (Lines 380 to 382)

Reviewer 3 Report

The paper is an interesting one particularly as it looks to see if progresses made in the area of women empowerment have been sustained over the years, and also to see ways to ensure this sustainability or to improve it. However, the following points must be considered. 

The abstract should tell clearly the main take home message

In the second paragraph of the introduction are statements of facts but unfortunately these are not backed with any literature. Must be backed with literature

In line 98-101, specifically, ‘Of particular note is the situation of women in Africa, where it is significant that there is a difference between rural and urban women around the continent. In rural areas, there are not enough resources for subsistence and the quality of those resources are not healthy at all’ this statement is subject to challenge and therefore must be supported with the relevant literature.

In line 305-306, the authors’ stated that ‘However, the level of empowerment is lower when it is they who occupy these positions of responsibility’. The reasons for this claim has to be discussed as it is an important point.

Check and rectify your citations from 317 to 330 as the authors mixed the actual citations with the order of citation numbers instead.

The discussion part has not been rigorous enough. For instance, the reason why the level of empowerment is lower when women are in positions of responsibilities compared to when it is men has to be discussed.

In lines 340-342, ‘Most of them are Spanish nationals and there are many permanent-discontinuous contracts. During the agricultural season they work and in the summer months they are unemployed. In this way they have an income every month of the year’. This appears contradictory. How do they earn income all year round if they are unemployed in the summer season?

In the conclusion, the recommendation does not seem to have much novelty as many studies have recommended the need to place more emphasis on the girl child’s education at all levels and in all study courses. I will however suggest that in addition to this recommendation, if reasons could be made for your findings on why there are differences in women empowerment when males are supervisors as against women, and also what accounts for the differences in satisfaction in the different years’ intervals, then you could find a new recommendations that can help sustain progresses.

In the second phase of the methodology, specifically line 134, 'Technological changes in the local production system will be analyzed'. The result of this step is however not apparent in the work. Perhaps such an analysis could enhance explanation and understanding on why there has been changes in the number of accidents and also the incidence rate.  

Author Response

Reviewer 3

The paper is an interesting one particularly as it looks to see if progresses made in the area of women empowerment have been sustained over the years, and also to see ways to ensure this sustainability or to improve it. However, the following points must be considered. 

Thank you for your comments. We really appreciate the effort you have made into this revision, and we are glad to know that you enjoyed the paper.

The abstract should tell clearly the main take home message

Thank you for your suggestion. We have inlcuded it (Lines 15 to 17).

In the second paragraph of the introduction are statements of facts but unfortunately these are not backed with any literature. Must be backed with literatura

We have included references to back those statements (References 2 and 3).

In line 98-101, specifically, ‘Of particular note is the situation of women in Africa, where it is significant that there is a difference between rural and urban women around the continent. In rural areas, there are not enough resources for subsistence and the quality of those resources are not healthy at all’ this statement is subject to challenge and therefore must be supported with the relevant literature.

We have included references to back those statements (Reference 16).

In line 305-306, the authors’ stated that ‘However, the level of empowerment is lower when it is they who occupy these positions of responsibility’. The reasons for this claim has to be discussed as it is an important point.

Thank you very much for your suggestion. We have included a discussion about it (Lines 314 to 319)

Check and rectify your citations from 317 to 330 as the authors mixed the actual citations with the order of citation numbers instead.

Sorry for the inconvenience. We have corrected it.

The discussion part has not been rigorous enough. For instance, the reason why the level of empowerment is lower when women are in positions of responsibilities compared to when it is men has to be discussed.

Thank you for your suggestion. We have included this issue in the discussion section (Lines 314 to 319) and we have expanded the discussion section (Lines 322 to 327)

In lines 340-342, ‘Most of them are Spanish nationals and there are many permanent-discontinuous contracts. During the agricultural season they work and in the summer months they are unemployed. In this way they have an income every month of the year’. This appears contradictory. How do they earn income all year round if they are unemployed in the summer season?

We agree with you. We have added information to clarify this issue (Lines 357 to 359)

In the conclusion, the recommendation does not seem to have much novelty as many studies have recommended the need to place more emphasis on the girl child’s education at all levels and in all study courses. I will however suggest that in addition to this recommendation, if reasons could be made for your findings on why there are differences in women empowerment when males are supervisors as against women, and also what accounts for the differences in satisfaction in the different years’ intervals, then you could find a new recommendations that can help sustain progresses.

You are right. We have included more recommendations in the conclusion section (Lines 367 to 369 and lines 376 to 378)

In the second phase of the methodology, specifically line 134, 'Technological changes in the local production system will be analyzed'. The result of this step is however not apparent in the work. Perhaps such an analysis could enhance explanation and understanding on why there has been changes in the number of accidents and also the incidence rate.

Thank you for this comment. We have addressed this topic in the discussion section (Lines 322 to 327)

Round 2

Reviewer 1 Report

After proofreading, the article posses a better flow for readers. I see much improvement in that.

The Authors include all underlined comments in the review. However, I have some minor comments.

General comments:

Abstract. Authors add additional sentences that cover the results of the article part. I don't know how to assess the part "women workers ….. are satisfied"? How to use that, satisfied in which field? Please be more accurate.

The introduction and literature review part I assessed as correct. I also noticed that Autorhs add a new up-to-date position in references.

In the previous version of the article methodology part, I noticed many gaps in the methodology of the study information. However, the authors add a few pieces of information, but in my opinion, it is not still fully characterized.

Page 4, line 154-156 – Please add what kind of level of turnover the companies need to exceed to be in your sample. Please add from which years you took the information about companies turnover.

Figure 5 has a different graphic style comparing to other figures in the article. Also, numbers - shares are not visible, so I cannot assess the scale of accidents presented there. It should be corrected.

Page 10, line 301 – how satisfaction was defined in the study. The literature part was not mentioned, but finally, it was underlined as w key results in the abstract. The authors wrote about that on page 11 line 322 and added 3 reference positions concerning that field. Please add the literature background on that in the literature part and add information about literature theory relation. Are there other articles that also concern the satisfaction of a job concern women in the fruit and vegetable sector in the food processing sector?

Discussion of the study was improved as well as the final conclusions. Authors add the part for further research and limitations as well as the contribution part.

Author Response

After proofreading, the article posses a better flow for readers. I see much improvement in that.

Thank you for your comments in the previous review. They have improved the quality of the paper.

The Authors include all underlined comments in the review. However, I have some minor comments.

General comments:

Abstract. Authors add additional sentences that cover the results of the article part. I don't know how to assess the part "women workers ….. are satisfied"? How to use that, satisfied in which field? Please be more accurate.

We agree with you. We have clarified the meaning of that satisfaction (Line 16)

The introduction and literature review part I assessed as correct. I also noticed that Autorhs add a new up-to-date position in references.

In the previous version of the article methodology part, I noticed many gaps in the methodology of the study information. However, the authors add a few pieces of information, but in my opinion, it is not still fully characterized.

Page 4, line 154-156 – Please add what kind of level of turnover the companies need to exceed to be in your sample. Please add from which years you took the information about companies turnover.

You are right. We have included this information about the selection process (Lines 164 to 166).

Figure 5 has a different graphic style comparing to other figures in the article. Also, numbers - shares are not visible, so I cannot assess the scale of accidents presented there. It should be corrected.

Sorry for the inconvenience. We have included the percentages in the figure 5.

Page 10, line 301 – how satisfaction was defined in the study. The literature part was not mentioned, but finally, it was underlined as w key results in the abstract. The authors wrote about that on page 11 line 322 and added 3 reference positions concerning that field. Please add the literature background on that in the literature part and add information about literature theory relation. Are there other articles that also concern the satisfaction of a job concern women in the fruit and vegetable sector in the food processing sector?

Thank you for this suggestion. We have included a paragraph in the literature review about satisfaction of women in the fruit and vegetable sector in the food processing sector (Lines 111 to 117 )

Discussion of the study was improved as well as the final conclusions. Authors add the part for further research and limitations as well as the contribution part.

Reviewer 3 Report

Thank you for trying your best to respond to the comments. In my view, the paper is now improved.

'This study's main contribution of this study lies in the extrapolation' in line 17-18 should be corrected.

Author Response

Thank you for trying your best to respond to the comments. In my view, the paper is now improved.

'This study's main contribution of this study lies in the extrapolation' in line 17-18 should be corrected.

Thank you for your comments in the previous review. The quality of the paper has improved. We have corrected the sentence, sorry for the inconvenience.